# Insights into the Mechanism of Action of *Helianthus annuus* (Sunflower) Seed Essential Oil in the Management of Type-2 Diabetes Mellitus Using Network Pharmacology and Molecular Docking Approaches

Athika Rampadarath, Fatai Oladunni Balogun and Saheed Sabiu *

Department of Biotechnology and Food Science, Durban University of Technology, Durban 4000, South Africa
* Correspondence: sabius@dut.ac.za

**Abstract:** Type-2 diabetes mellitus (T2D) is one of the leading non-communicable diseases of global concern. Knowing the exact mechanism of action of available antidiabetic agents, particularly natural products, may assist in providing effective therapeutic solutions. The antidiabetic action of *Helianthus annuus* (sunflower) seed has been established; however, the molecular mechanism of action, especially the essential oil, is lacking. The study explored network pharmacology and molecular docking studies to determine the active phytoconstituents, signaling pathways, and probable therapeutic targets to determine the antidiabetic potential of sunflower seed essential oil. Preliminary analysis established 23 target genes with 15 phytoconstituents involved in T2D which all passed Lipinski's rule of five with no violation. Three pathways were proposed by KEGG analysis as therapeutic targets for T2D development with PPAR as the major route affecting PPARA, FABP4, PPARD, PPARG, and CPT2 genes. Molecular docking investigation confirmed the effectiveness of active SSEO compounds against the identified genes (targets) and established phylloquinone, linoleic acid, tricosylic acid, and lignoceric acid as the probable drug candidates that could offer laudable therapeutic effects in an effort towards T2D management. Thereby, we present an insight toward understanding the mechanism of the antidiabetic action of sunflower seeds via the stimulation of glucose to enhance insulin release.

**Keywords:** sunflower; *Helianthus annuus*; molecular docking; network pharmacology; type-2 diabetes mellitus; essential oil





## 1. Introduction

Diabetes mellitus is a chronic metabolic disorder caused by the inability of the pancreas to secrete insulin or the insensitivity of the produced insulin to absorb the available glucose, leading to hyperglycemia [1]. While this health problem may be broadly classified into two groups, i.e., insulin-dependent (type 1) and non-insulin-dependent (type 2, accounting for 90% of diabetes cases), there is a third classification known as gestational diabetes that arises when hormones due to pregnancy escalate the glucose level in the blood [2]. Sadly, the economic burden (and deaths) accompanying the global prevalence of type-2 diabetes mellitus (T2D) particularly among the working population continues to increase [3]. In fact, as of 2013, 380 million people were reported to be living with diabetes [4]; sadly, in 2014, the figure rose to 422 million (with 8.5% of all adults being diabetic) [2] and the overall estimation by 2035 reached more than 590 million [3,4]. It is worthy of note that in 2021, around 535 million adults (20–79 years) were reported to have been suffering from diabetes. Unfortunately, the figure (535 million) is approaching the 2035 projection of 590 million, implying the grievous consequences and burden of the disease. In fact, 1 in every 10 individuals is reported to be suffering from T2D globally, and this is predicted to rise to 643 million by 2030 and 783 million by 2045 [5]. The approach toward the management

of T2D varies, including non-pharmacological methods (involving regular exercise and different dietary regimens) and pharmacologically directed methods which explore the use of oral hypoglycemics (OHAs) including sulfonyl ureas (glibenclamide), biguanides (metformin), meglitinide (repaglinide), thiazolidinedione (rosiglitazone), alpha-glucosidase inhibitors (acarbose), dipeptidyl peptidase inhibitors (sitagliptin). Additionally, apart from the aforementioned therapeutic interventions, the functionalization or the incorporation of amino or azido groups to sugars (imino sugars or sugar derivatives) is another notable approach explored in diabetes management [6–8]. While the ultimate goal for the usage of these agents or modifications in T2D therapy is centered towards normalizing the glucose level in the blood, the unavailability, inaccessibility (to diabetic sufferers), non-portable nature, and side effects derived from them have inspired an alternative choice involving the use of medicinal plants or natural products judged to lack many of the disadvantages attributed to synthetic drugs [9,10].

*Helianthus annuus* is a member of the family Asteraceae [11] endemic to North America, but is now widely distributed in many continents (of the world) including Africa. It is an economically important oilseed crop with global cultivation [12] and is in fact considered the third best in production after soybean and rapeseed crops [13], with its worldwide cultivation exceeding 56.97 million tonnes in 2021 [13,14]. The seed of the plant is embraced for several nutritional purposes not limited to its use in the making of snacks and the preparation of (vegetable) delicacies. While these seeds are rich in oil (36–50%) [15] and contain largely unsaturated fatty acids [16,17], the plant and its parts are endowed with abundant pharmacological potential [18], including antioxidant [12], antimicrobial [19–21], and cytotoxic [21] effects attributed to its metabolites (flavonoids, phenolic acids, tocopherol, saponins, alkaloids, tannins, and terpenes) and/or several phytoconstituents [20]. Additionally, while the quantitative determination of the chemical contents of the various cultivars of the seeds has revealed 23 saturated and unsaturated fatty acids with linoleic acid being the most abundant [22,23], the antidiabetic effects of the plant seed have been established in a few reports [24] as, for example, recently reviewed by Rehman et al. [25], though no study has reported the antidiabetic potential of the oilseed of the plant. Moreover, since the quality and therapeutic effectiveness are known as factors which determine the oilseed's ability to lower blood cholesterol and reducing the risk of cardiovascular diseases, among others, it is important to study the therapeutic role the inherent phytoconstituents play in exhibiting or being responsible for these effects, particularly against T2D as well as the molecular mechanisms underlying their action. Notwithstanding the aforementioned, the plant (seed oil) has found applications in clinical trials (involving animals and humans). Typically, it is reported to cause a reduction in bacterial infection in pre-term babies following topical application in Egypt [26]. Similarly, as an emollient therapy, its use has been clinically submitted in reducing skin barrier integrity and risk of infection and mortality in very-low-birth-weight (<1.5 g) infants in a randomized controlled trial in Uttar Pradesh, India [27]. While reports on its application in the treatment of tinea pedis are also available [28], reports of clinical trials in the healing process of wounds experimentally induced in horses and as a pro-inflammatory agent in fish have also been established [29,30].

In the past, approaches used in the development or discovery of most conventional drugs have been tailored towards specific protein targets; however, in recent times, the complexities between the active ingredients and elicited therapeutic effects are studied through a complex biological network system (network pharmacology) by medicinal plant researchers to derive an effective drug [31,32]. Network pharmacology (NP) is a computer-based-biological research technology that studies the interplay between drugs and disease targets with the sole goal of identifying the bioactive constituents and diseases' therapeutic targets [33]. The use of NP in this study affords the opportunity to discover potential therapeutically viable bioactive metabolites and the signaling pathways to be targeted while using sunflower seed essential oil against T2D. The approach allows for multidrug-multi-target/genes exploration against T2D. Molecular docking addresses the binding

relationship or affinities of potential compounds (sunflower seed essential oil, SSEO) with targets (protein/disease) for this study. Both computational methods are ideal approaches for screening potential candidates that could be taken further into experimental (in vitro and in vivo) studies in an effort toward drug development. However sadly, the use of NP to X-ray the relationship between phytoconstituents of sunflower seed essential oil and T2D targets has not, to the best of our knowledge, been reported. Hence, with this technology and/ or network, the therapeutically active components of sunflower for the management of diabetes mellitus are aimed to be discovered.

## 2. Materials and Methods

### 2.1. Materials

Source of Seeds

Sunflower seed sample pre-treated with pesticides was sourced from the Agricultural Research Council (ARC) Grain Crop Institute, Potchefstroom, South Africa.

### 2.2. Methods

2.2.1. Seed Preparation, Extraction, and Chromatographic Analysis

Following the preliminary processes of washing (with distilled water) and cleaning (to rid of germs), seed coat removal, drying (at 37 °C for 24 h), and grinding (mortar and pestle) the seeds, the powdered seeds were subjected to Soxhlet extraction [34,35] using petroleum ether of 250 mL (40/60 strength) as a solvent of extraction to obtain the oil. Two (2) ml (2:1 chloroform: methanol) was added to ca. 100 mg oil, vortexed, sonicated (25 °C/30 min), and centrifuged (3000 rpm for 1 min). The obtained bottom layer was dried, reconstituted (with methyl tert-butyl ether and trimethyl sulfonium hydroxide; 10:3), and vortexed. Exactly 1 μL of the derivatized mixture was injected in a 5:1 ratio onto the gas chromatography (6890N, Agilent Technologies, Santa Clara, CA, USA)–flame ionization detector equipment (GC–FID) for further chromatographic separation.

The separation of the fatty acid methyl esters (FAMEs) was performed [36] on a polar capillary column (RT-2560, Specifications: 100 m, 0.25 mm ID, 0.20 μm film thickness) (Restek, Bellefonte, PA, USA). Helium was adopted as the carrier gas at a 1 mL/min flow rate. The injector temperature was maintained at 240 °C. The oven temperature was programmed to 60 °C for 1 min, ramped to 120 °C at a rate of 8 °C/min for 1 min, followed by a ramping rate of 1.5 °C/min to 245 °C for 1 min, and a final ramp up to 250 °C at a rate of 20 °C/min for 2 min. Based on this, the identification of various fatty acids was determined by their retention times or elution (from the column based on the respective carbon numbers or atoms) measurement through the detector.

2.2.2. Screening of Active Compounds, Drug Therapeutics, and Disease Targets

The 15 identified compounds (with retention times) from the GC–FID/FAMEs analyses (Supplementary Figures S1 and S2) and Dr. Duke Phytochemical and Ethnobotanical database (https://phytochem.nal.usda.gov/phytochem/plants/show/1011?et=; obtained 15 September 2022) were subjected to SwissADME (http://swissadme.ch/index.php; accessed on 15 September 2022) [37] to predict their absorption, distribution, metabolism, and excretion properties in line with Lipinski's rule of five [38]. For the data mining, the canonical SMILES (simplified molecular input line entry system) information generated from the pasting of the compound names on the PubChem website (https://pubchem.ncbi.nlm.nih.gov/, obtained on 15 September 2022), was analyzed through two databases; Similarity Ensemble Approach (SEA, https://sea.bkslab.org/ obtained on 15 September 2022) and Swiss Target Prediction (STP, http://www.swisstargetprediction.ch/ assessed on 15 September 2022) to identify genes associated with sunflower seeds essential oil (SSEO). Similarly, the GeneCards (https://www.genecards.org, used 15 September 2022) [39] and DisGeNET (https://www.disgenet.org/search, accessed 15 September 2022) databases were explored for the generation of genes associated with T2D. Using "type-2 diabetes mellitus" as a keyword, the disease targets were obtained in the DisGeNET database [40] (with

Score gda > median) and (Relevance score > 2) in the Gene Cards database as screening conditions. Drug and disease targets were selected as candidates, and a corresponding Venn diagram was generated by Venny 2.1 (https://bioinfogp.cnb.csis.es/tools/venny/ obtained on 15 September 2022) to depict the intersection of genes between the SSEO constituents and T2D.

### 2.2.3. Protein–Protein Interaction (PPI) Network Construction

The STRING (Search Tool for the Retrieval of Interacting Genes/Proteins) 11.0 data platform (https://cn.string-db.org/; accessed on 10 September, 2022) and the Cytospace (version 3.9.1) application were used to explore the functional link between proteins based on their connections between the genomes [41]. Briefly, for the network analysis, the data generated from the STRING database downloaded in TSV format was imported into the Cytoscape software (https://cytoscape.org/download.html; V3.7.2, Seattle, WA, USA assessed on 25 September 2022). The software classifies all the networks and makes the important or integral genes to be identified following a degree algorithm using the below expression. Finally, the Cytoscape software helped with the visualization of network analysis of crucial gene clusters.

$$Deg\ (v) = |\ N(v)$$

where $N(v)$ is the node neighbor and $v$ represents each node's neighbors.

### 2.2.4. Kyoto Encyclopaedia of Genes and Genomes (KEGG) Pathways Enrichment Analyses

To determine the biological connection of key crucial genes (CG) involved in T2D, the (CG) obtained from the PPI analysis was imported into the database for annotation, visualization, and integrated discovery (DAVID, version 6.8 [42], assessed through https://david.ncifcrf.gov/tools.jsp, on 25 September 2022). The pathway enrichment had a benchmark of $p < 0.05$ and the false discovery rate error control method had results expressed as a value, 'Q'. The KEGG analysis assisted with the signaling pathway of the CG clusters while the micro-biographic mapping platform (http://www.bioinformatics.com.cn/, obtained on 25 September 2022) and Cytoscape software were employed for results visualization with the aid of bubble plot.

### 2.2.5. Molecular Docking

The 3D structures of the most significantly enriched genes (PPARA, PPARG, PPARD, CPT2, and FABP4) with PDB IDs 3ET1, 6TSG, 5Y7X, 2FW3, and 5D4A, respectively, linked to the most enriched (PPAR) signaling pathway connected to T2D were obtained from the RCSB PDB database (https://www.rcsb.org/, accessed on 3 October 2022). The ligand-bound selected targets were chosen based on their low resolutions which were between 1.5 and 2.5 Å. Before molecular docking, the targets were prepared/optimized using UCFS Chimera software v. 1.14 by removing water molecules, heteroatoms, native ligands, and non-standard amino acids before the addition of missing side chains [43]. However, for the ligand preparation, the SDF 3D format of the constituents of SSEO and the standards (metformin (reference) and rosiglitazone (antagonist)) were downloaded from PubChem (https://pubchem.ncbi.nlm.nih.gov/, accessed on 3 October 2022), and subsequently optimized and energy minimized in UCFS Chimera software v. 1.14 through the inclusion of non-polar hydrogen atoms and Gasteiger charges and saved in PDB format as prepared ligand molecules [44]. The prepared proteins and ligands were docked using AutoDock Vina software (V1.1.2, La Jolla, CA, USA), a plugin program for molecular docking [45]. The grid box was created and spaced by 1 Å while the defined sizes stretch to the x, y, z directions in each case. Thereafter, the docking affinity scores and the various interactions arising from the formation of each complex were generated. However, to prevent a pseudo-binding pose or conformation, a confirmation of the docking protocol was carried out; this was achieved by measuring the root mean square deviation of the docked ligand from the

binding pocket containing the native inhibitors within the co-crystalized structures of the five studied receptors (3ET1, 6TSG, 5Y7X, 2FW3, and 5D4A) after optimal superimposition. The RMSD scores obtained (0.5 Å) between the native ligands and the docked compounds within the 3D structures of the receptors revealed binding orientations as presented in Appendix A, confirming the validation of the explored protocol.

## 3. Results

### 3.1. Compounds Identification and ADME Properties Screening

Fifteen compounds (capric acid, caproic acid, caprylic acid, lauric acid, myristic acid, palmitic acid, pentadecyclic acid, stearic acid, oleic acid, linoleic acid, arachidic acid, behemic acid, tricosylic acid, lignoceric acid, and phylloquinone) were identified from GC–FAME analysis and data mining. The 15 compounds were selected based on passing Lipinski's rule of five with no violation (Table 1).

**Table 1.** GCMS–FAMEs phytoconstituents identification from sunflower seeds essential oil and Lipinski properties evaluation.

| S/N | Compound Names | Lipinski's Rule Remarks | No of Violations |
| --- | --- | --- | --- |
| 1 | Capric acid | Yes | None |
| 2 | Caproic acid | Yes | None |
| 3 | Caprylic acid | Yes | None |
| 4 | Lauric acid | Yes | None |
| 5 | Myristic acid | Yes | None |
| 6 | Palmitic acid | Yes | None |
| 7 | Pentadecyclic acid | Yes | None |
| 8 | Stearic acid | Yes | None |
| 9 | Oleic acid | Yes | None |
| 10 | Linoleic acid | Yes | None |
| 11 | Arachidic acid | Yes | None |
| 12 | Behenic acid | Yes | None |
| 13 | Tricosylic acid | Yes | None |
| 14 | Lignoceric acid | Yes | None |
| 15 | Phylloquinone | Yes | None |

### 3.2. Screening of Active Compounds and Their Targets

One hundred and eighty-two (182) and 365 genes associated with the SSEO target (following duplicate removal) were retrieved from Similarity Ensemble Approach (SEA) and Swiss Target Prediction (STP) databases, respectively. However, only 23 (4.4%) significant genes associated with SSEO were common to both databases, as depicted in Figure 1A.

### 3.3. Screening of T2D Disease Targets, Drugs, and Disease Candidates

The GeneCards database recorded 13,395 genes associated with T2D targets while genes associated with targets of T2D retrieved from the DisGeNet database numbered 3134 with an overlap of 2603 genes common to both databases (Figure 1B). However, since all the genes could not be worked with, cross-matching genes of the active compounds of SSEO from the two databases (23) against the overlapping genes from the disease (T2D) targets databases (2603) (Figure 2) resulted in 17 (0.7%) genes common to SSEO and T2D targets.

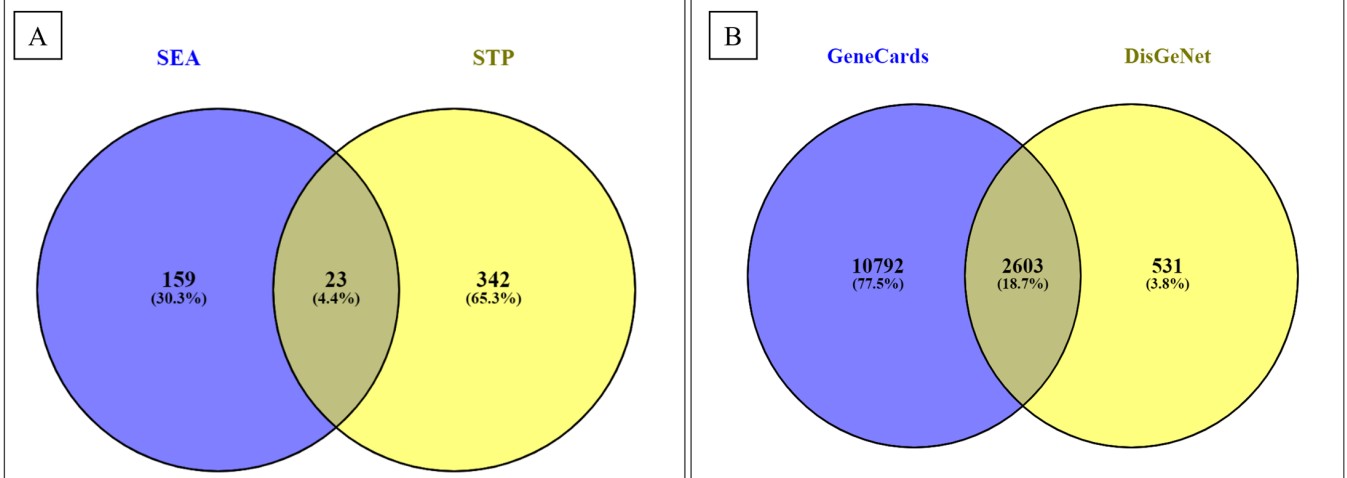

**Figure 1.** Venn diagram depicting the overlapping genes (**A**) of active compounds of sunflower seeds essential oil (SSEO) from two databases (Similarity Ensemble Approach and Swiss Target Prediction) and (**B**) associated with type-2 diabetes mellitus targets (from GeneCards and DisGeNet databases).

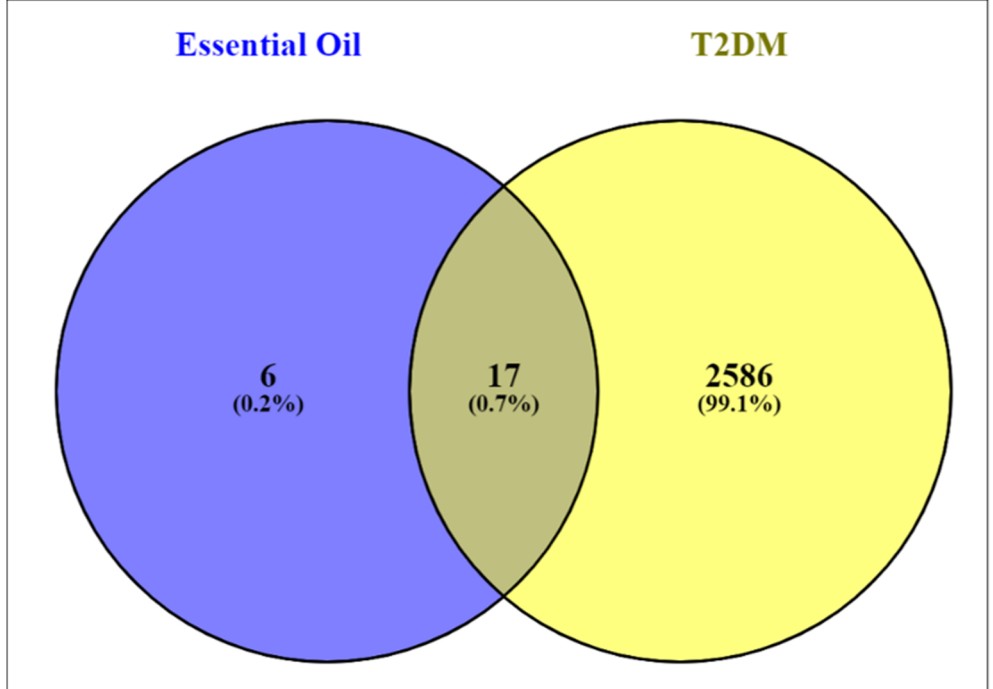

**Figure 2.** Venn diagram showing the common genes between sunflower seed essential oil and type-2 diabetes mellitus.

### 3.4. Protein–Protein Interaction Network

The 17 SSEO–T2D overlapping genes presented a network depicting 17 nodes (containing PPARD, FABP4, PTPRC, NODI, LPAR1, GPR35, PPARG, PPAR, TRPV1, CNR1, FFAR4, GSTK1, TOP1, RARB, FOLH1, GPR35, LPAR1, and CPT2) among the involved proteins and 24 edges connecting each respective nodes and achieved in a PPI construction network made from STRING and Cytoscape (Figure 3). The edges were taken as the number of degrees for each target, indicating targets with the highest number of degrees as the network's best or leading target.

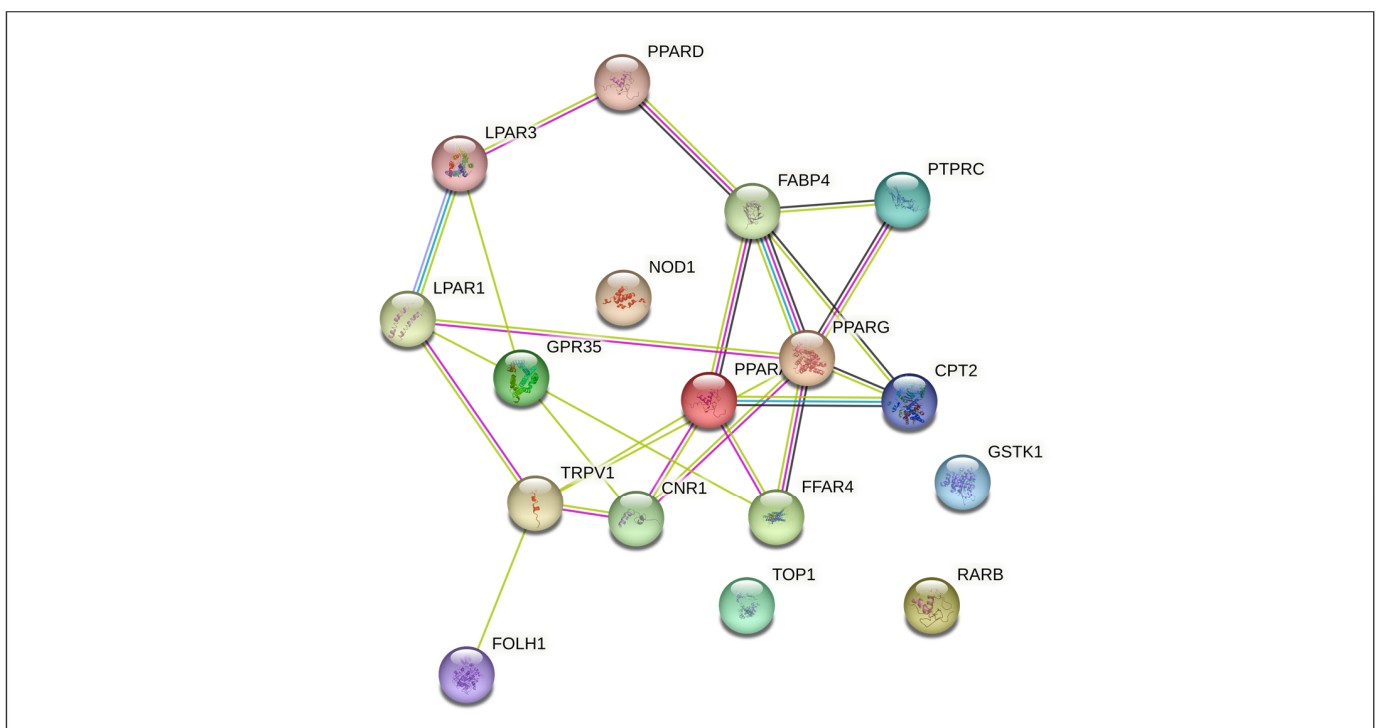

**Figure 3.** Protein–protein interaction network construction between sunflower seed essential oil (SSEO)-type-2 diabetes mellitus (T2D) (17 nodes and 24 edges were constructed from the network).

### 3.5. KEGG Enrichment Analysis

The enrichment analysis on the 17 prospective gene targets produced three KEGG signaling pathways (Table 2) implicated in T2D achieved at a threshold of $p < 0.05$. Based on further analysis and the generated bubble plot, while each of the three identified pathways is endowed with five genes each, the PPAR was observed as the best pathway owing to its lowest false discovery rate value ($2.01 \times 10^{-6}$) and characterized by FABP4, PPARG, PPARD, CPT2, and PPARA exhibiting, different interactions with the SSEO compounds (Figure 4). However, out of the 5 identified genes, PPARA was discovered as the most important (highest degrees) based on its interactions with all the 15 SSEO compounds (Figure 5A) followed by FABP4 and PPARD interacting with 14 (except phylloquinone) out of the 15 SSEO compounds (Figure 5B,C, respectively). The lowest were PPARG (13 compound interactions excluding phylloquinone and linoleic acid) (Figure 5D) and CPT2 with 12 compounds (except phylloquinone, linoleic, and oleic acids) (Figure 5E).

**Table 2.** Target genes in KEGG enrichment analysis of SSEO associated with T2D.

| Term ID | Pathways | No of Genes | False Discovery Rate | Genes |
|---------|----------|-------------|----------------------|-------|
| hsa03320 | PPAR signaling pathway | 5 | $2.01 \times 10^{-6}$ | FABP4, PPARG, PPARD, CPT2, PPARA |
| hsa04080 | Neuroactive ligand-receptor interaction | 5 | $1.20 \times 10^{-3}$ | CNR1, LPAR1, LPAR3, GPR35, TRPV1 |
| hsa05200 | Pathways in cancer | 5 | $7.00 \times 10^{-3}$ | PPARG, PPARD, RARB, LPAR1, LPAR3 |

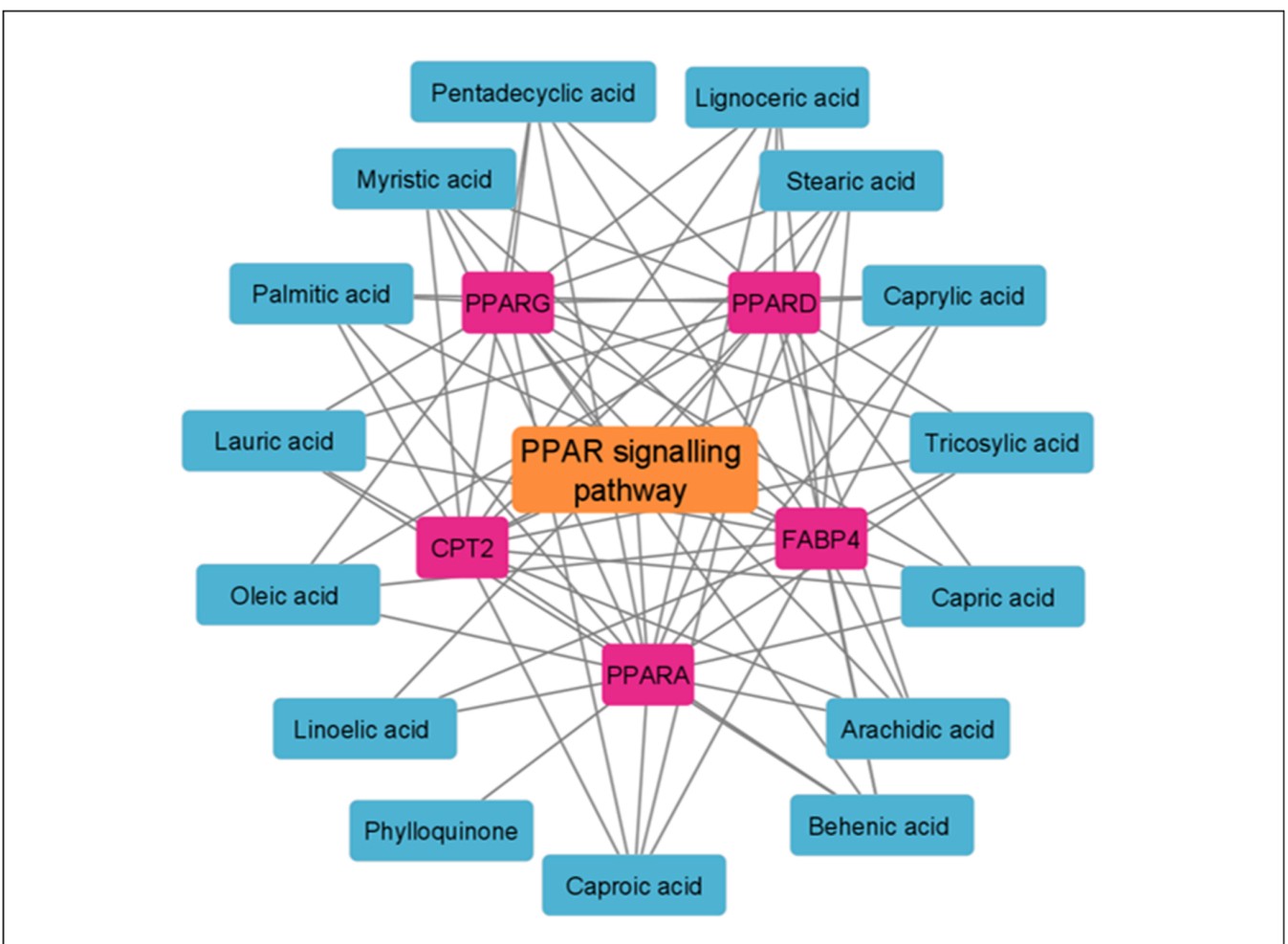

**Figure 4.** Gene–gene interactions of five genes linked to the PPAR signaling pathway of the bioactive constituents of sunflower seeds essential oil against T2D.

*3.6. Molecular Docking*

The outcome of the docking evaluation arising from the interaction of SSEO compounds and standards with the respective active sites of the five genes (PPARA, FABP4, PPARG, PPARD, and CPT2) is presented in Supplementary Table S1, while the results for the top five compounds/standard against each gene are shown in Table 3. Typically, against PPARA, phylloquinone (−9.2 kcal/mol) had the best docking score when compared with the other compounds and standards, i.e., rosiglitazone (−8.6 kcal/mol) and metformin (−5.2 kcal/mol). The degree or pattern of affinity was phylloquinone > rosiglitazone > lignoceric acid > tricosylic acid > behenic acid > linoleic acid > metformin. However, with the other four receptors, the inferiority of the complexes with each of the top five compounds compared with rosiglitazone was established though the compounds, which all revealed the most negative docking scores relative to metformin (reference drug), indicating their better binding affinities (Table 3). Additionally, the summary of the results of the interaction between the respective ligands (SSEO compounds and standards) and targets are shown in Table 4, while the comprehensive report on the interactions is presented in Supplementary Figure S3.

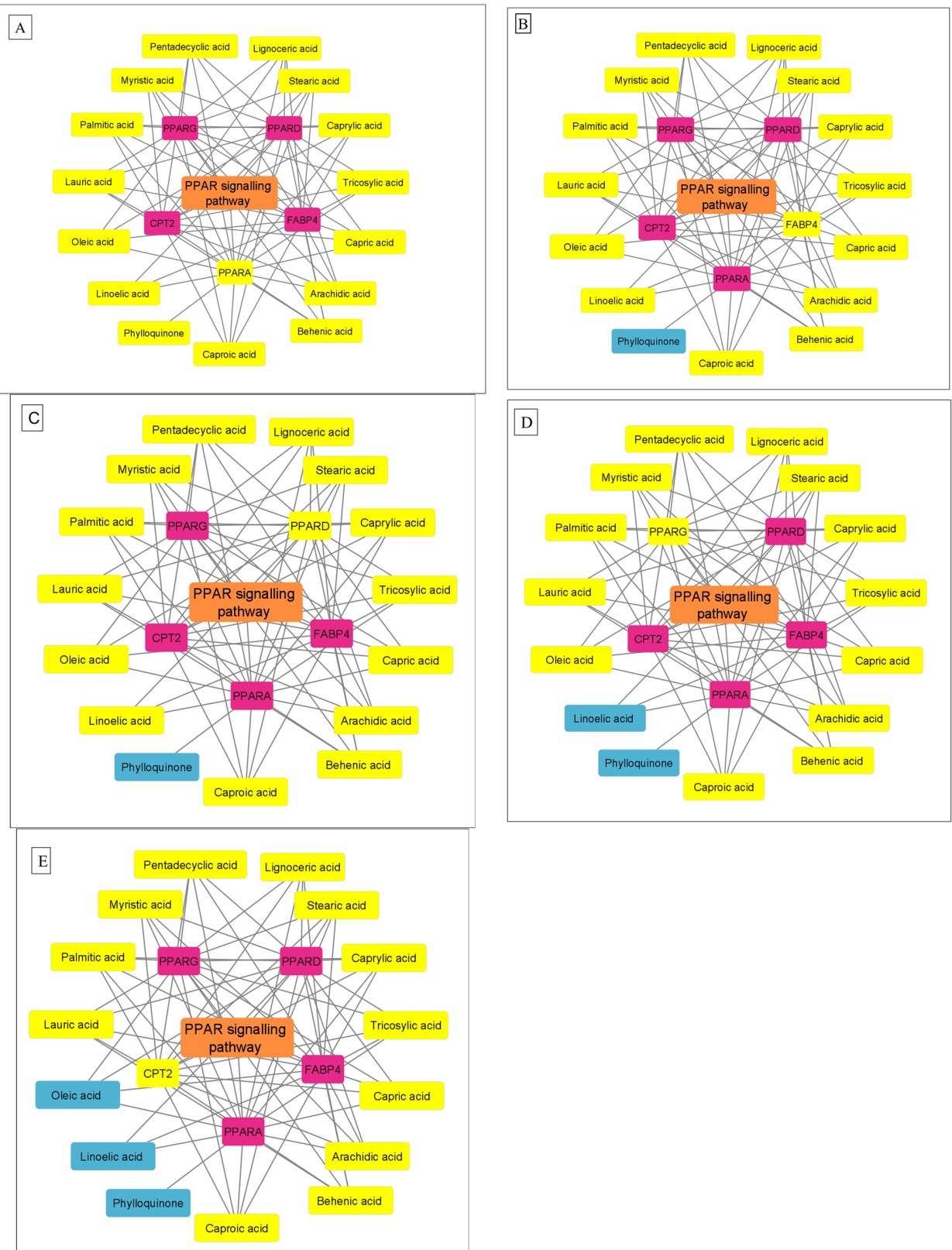

**Figure 5.** (**A**) PPARA gene (yellow node) interactions with 15 bioactive constituents of sunflower seeds essential oil (SSEO) against T2D; (**B**) FABP4 gene (yellow node) interaction with 14 SSEO compounds; (**C**) PPARD (yellow node) interaction with 14 compounds; (**D**) PPARG (yellow node) with 13 compounds and (**E**) CPT2 (yellow node) interactions with 12 compounds.

**Table 3.** Docking score (kcal/mol) of the top five sunflower seed essential oil compounds and targets in the PPAR pathway.

| S/N | Compounds/Standards | Docking Scores (kcal/mol) | | | | |
|---|---|---|---|---|---|---|
| | | **PPARA** | **FABP4** | **PPARD** | **PPARG** | **CPT2** |
| 1 | Linoleic acid | −6.3 | −6.3 | NA | NA | NA |
| 2 | Lignoceric acid | −6.7 | −6.0 | NA | −6.1 | −5.7 |
| 3 | Behenic acid | −6.5 | −5.9 | NA | NA | −6.1 |
| 4 | Phylloquinone | −9.2 | NA | NA | NA | NA |
| 5 | Tricosylic acid | −6.6 | −6.0 | −7.1 | −5.7 | −5.7 |
| 6 | Arachidic acid | NA | −5.9 | −6.1 | −5.8 | −6.2 |
| 7 | Pentadecyclic acid | NA | NA | −6.3 | NA | NA |
| 8 | Stearic acid | NA | NA | −6.3 | −5.8 | −6.3 |
| 9 | Oleic acid | NA | NA | −6.4 | NA | NA |
| 10 | Palmitic acid | NA | NA | NA | −5.8 | NA |
| 11 | Rosiglitazone | −8.6 | −8.3 | −8.7 | −8.2 | −9.4 |
| 12 | Metformin | −5.2 | −4.5 | −5.2 | −4.9 | −5.3 |

NA: Not applicable.

**Table 4.** Identified interactions between the top five SSEO phytoconstituents and the PPAR signaling genes.

| Complex | Number of Interactions | Number of H-Bonds and Interaction Residues | Number of van der Waal Forces and Interaction Residues | Other Important Interactions and Residues |
|---|---|---|---|---|
| PPARA–phylloquinone | 25 | - | 16 (Gln277, Met330, Met320, Thr279, Gly335, Val332, Tyr334, Leu460, Tyr464, His440, Tyr314, Ser280, Thr283, Asn219, Phe318, Lys358) | 9 (Cys276, Met355, Ile354, Ile317, Leu321, Val324, Met220, Leu331, Phe273) |
| PPARA–lignoceric acid | 22 | 2 (His440, Tyr464) | 14 (Tyr314, Lys358, Met355, Cys276, Gln277, Ser280, Thr283, Met320, Asn219, Thr279, Tyr334, Gly335, Val332, Leu331) | 6 (Phe318, Ile217, Leu321, Met220, Met330, Val324) |
| PPARA–behenic acid | 21 | - | 12 (Phe218, Met320, Val324, Thr283, Thr279, Tyr314, Phe318, Ile354, Gln277, Asn219, Met220, Ser280) | 9 (Leu321, Ile317, His440, Cys226, Phe273, Leu458, Leu460, Val444, Tyr464) |
| PPARA–tricosylic acid | 21 | 3 (Gln277, Ser280, Tyr464) | 9 (Val444, Phe318, Thr279, Val332, Tyr334, Leu331, Thr283, Tyr314, Leu460) | 9 (Ile354, Phe273, Lys276, His440, Leu321, Ile317, Met320, Val324, Met220) |
| PPARA–linoleic acid | 14 | 1 (Cys276) | 9 (Val332, Leu331, Asn219, Glu286, Phe218, Thr283, Thr279, Ile317, Ser280) | 4 (Met220, Met320, Val324, Leu321) |
| PPARA–rosiglitazone | 20 | 1 (Ser280) | 14 (Asn219, Phe218, Met220, Ser323, Val324, Asn221, Met320, Thr279, Phe318, Lys358, Ile354, Gln277, Thr283, Met355) | 5 (Ile317, Leu321, His440, Cys276, Phe273) |
| PPARA–metformin | 13 | 2 (Tyr214, Thr283) | 9 (Lys 222, Thr279, Val324, Ser323, Asn221, Met220, Met220, Asn219, Phe218) | 2 (Glu286, Asp372) |
| FABP4–linoleic acid | 22 | 1 (Arg126) | 12 (Gln95, Thr74, Arg78, Val25, Asp76, Lys58, Ser53, Ser55, Cys117, Val115, Tyr128, Arg106) | 9 (Tyr19, Phe16, Met20, Ala75, Ala33, Ala36, Pro38, Phe57, Ile104) |
| FABP4–lignoceric acid | 8 | 1 (Thr29) | 4 (Met35, Phe27, Phe57, Lys58) | 3 (Lys31, Val32, Ala28) |
| FABP4–aradichic acid | 10 | 2 (Ala75, Asp76) | 4 (Asp77, Thr29, Lys58, Phe27) | 4 (Val32, Phe57, Ala28, Lys31) |
| FABP4–behenic acid | 22 | 1 (Arg106) | 11 (Arg78, Asp76, Ser55, Lys58, Ser53, Arg126, Thr60, Ile104, Met40, Val115, Tyr128) | 10 (Val25, Val23, Tyr19, Met20, Phe57, Ala33, Ala36, Ala75, Phe16, Pro38) |
| FABP4–tricosylic acid | 24 | 2 (Ala75, Thr24) | 11 (Glu72, Thr60, Asp76, Arg78, Arg126, Ser53, Lys58, Ser55, Val25, Tyr19, Arg106) | 11 (Met20, Ala36, Pro38, Phe51, Ala33, The16, Tyr128, Cys117, Ile104, Met40, Val115) |

<div align="center">

**Table 4.** *Cont.*

</div>

| Complex | Number of Interactions | Number of H-Bonds and Interaction Residues | Number of van der Waal Forces and Interaction Residues | Other Important Interactions and Residues |
|---|---|---|---|---|
| FABP4–rosiglitazone | 22 | 2 (Arg106, Ser53) | 14 (Val23, Arg78, Tyr19, Val115, Tyr128, Ser55, Val25, Met20, Asp76, Gln95, Ala75, Ala33, Phe57, Thr60) | 6 (Phe16, Cys117, Ile104, Arg126, Pro38, Ala36) |
| FABP4–metformin | 10 | 1 (Tyr128) | 7 (Met40, Pro38, Ala75, Ala36, Phe57, Phe16, Ala33) | 2 (Arg126, Ser53) |
| PPARD–tricosylic acid | 25 | 1 (His287) | 9 (Phe316, Trp228, Leu317, Thr252, Phe291, Phe246, Gln250, Leu433, Met417) | 15 (Leu219, Ile213, Val312, Arg218, Val245, Leu323, Val305, Leu294, Cys249, Ile322, Ile328, Tyr437, Thr253, Lys331, His413) |
| PPARD–oleic acid | 19 | 1 (Thr252) | 4 (Trp228, Thr253, Ile290, Leu317) | 13 (Phe316, Arg248, Val305, Leu294, Leu303, Cys249, Ile328, Lys331, Phe291, Val312, Val245, Leu219, Ile213) |
| PPARD–stearic acid | 20 | 6 (Thr253, Gln250, Phe246, Phe291, His244, Phe316), 1 (His413) | 1 (His413) | 13 (Leu219, Arg248, Val245, Val312, Leu303, Cys249, Lys331, Ile328, Leu317, Leu294, Ile213, Val305, Trp228) |
| PPARD–arachidic acid | 23 | 2 (His413, Thr253) | 8 (Trp228, Phe316, Ile290, Tyr437, His287, Met417, Leu433, Gln250) | 13 (Lys331, Phe291, Leu294, Ile328, Val305, Val312, Ile213, Cys249, Arg248, Leu219, Val245, Leu317, Leu303) |
| PPARD—pentadecylic acid | 19 | - | 7 (His287, His413, Ile290, Thr253, Phe291, Phe316, Trp228) | 12 (Ile328, Val305, Val245, Leu219, Leu294, Leu303, Arg248, Lys331, Leu317, Val312, Ile213, Cys249) |
| PPARD—rosiglitazone | 19 | 3 (Val245, His244, Ile327) | 11 (Leu219, Arg248, Ile328, Lys331, Met417, Phe246, Thr253, Phe291, Val305, Ile213, Trp228) | 5 (Val312, Leu317, Leu303, Cys249, His413) |
| PPARD–metformin | 9 | 2 (Met293, Thr256) | 5 (Tyr186, Asn191, Met192, Ile297, Ser296) | 2 (Glu259, Phe190) |
| PPARG–lignoceric acid | 24 | 1 (Gln286) | 15 (Ser289, Val339, Leu340, Ile341, Glu295, Phe226, Pro227, Phe287, Phe363, His449, Tyr327, Met364, Ser342, Lys367, Leu453) | 8 (Cys285, Arg288, Leu333, Ala292, Met329, Ile326, Leu228, Leu330) |
| PPARG–palmitic acid | 20 | 1 (Gln286) | 12 (Leu228, Glu295, Ile296, Ser289, Tyr322, His323, Tyr473, His449, Leu453, Lys367, Cys285, Met364) | 7 (Pro227, Phe226, Met329, Arg288, Ile326, Ala292, Leu330) |
| PPARG–stearic acid | 18 | 2 (Gln286, His449) | 6 (Glu295, Lys367, Phe363, Ser280, Tyr322, Ile296) | 10 (Phe226, Met329, Ala297, Ile326, Arg288, Glu330, Pro227, Leu228, Cys285, Met364) |
| PPARG–arachidic acid | 16 | 1 (Glu343) | 9 (Leu228, Leu340, Ser342, Val339, Met364, Leu333, Ser289, Glu295, Tyr327) | 6 (Cys285, Ile341, Leu330, Ile326, Ala292, Met329) |
| PPARG–tricosylic acid | 18 | 1 (Glu295) | 7 (Glu343, Ser342, Pro227, Phe226, Leu340, Ser289, Ile325) | 10 (Leu228, Leu333, Arg288, Leu330, Ala292, Met329, Ile326, Ile296, Cys285, Ile341) |
| PPARG–rosiglitazone | 17 | 6 (Leu228, Arg288, Pro227, Phe226, Ser332, Cys285) | 6 (Glu295, Ile341, Thr229, Met329, Leu333, Ser289, Ile341) | 5 (Ile326, Ala292, Leu330, Val339, Met364) |
| PPARG–metformin | 12 | 2 (Leu228, Ile326) | 9 (Pro227, Phe226, Arg288, Met329, Leu333, Ala292, Ser332, Ile296, Leu330) | 1 (Glu295) |
| CPT2–stearic acid | 16 | 3 (Thr499, Ser488, Tyr120) | 7 (Asp376, Trp116, Arg554, Ser588, Asn585, Met119, Asn130), | 6 (Tyr486, Val605, His372, Phe131, Phe602, Pro133) |
| CPT2–tricosylic acid | 22 | 3 (Ser488, Tyr486, Asp376) | 9 (Arg554, Trp116, Thr499, Met119, Gly377, Ser590, Asn585, Ser588, Asn130) | 9 (Val605, Phe131, Phe602, Met135, His372, Tyr120, Phe370, Pro133, Leu592) |

**Table 4.** *Cont.*

| Complex | Number of Interactions | Number of H-Bonds and Interaction Residues | Number of van der Waal Forces and Interaction Residues | Other Important Interactions and Residues |
|---|---|---|---|---|
| CPT2–lignoceric acid | 15 | 2 (Val175, Glu174) | 3 (Ser490, Val378, Arg382) | 10 (Phe176, Leu212, Ala547, Tyr210, Ala493, Phe494, Met548, Pro50, Trp201, Tyr205) |
| CPT2–arachidic acid | 16 | 2 (Ser498, Tyr486) | 9 (Trp116, Thr499, Arg554, Met119, Asn130, Ser590, Phe370, Ty120, Ser588) | 5 (Val605, Phe131, Phe602, His372, Pro133) |
| CPT2–behenic acid | 17 | 3 (Tyr486, Ser488, Asp376), | 8 (Asn585, Ser588, Ser590, Trp116, Asn130, Tyr120, Thr499, Arg554) | 6 (Val605, Phe131, Met119, Phe602, His372, Pro133) |
| CPT2–rosiglitazone | 21 | 1 (Ser590) | 15 (Val597, Ala615, Leu599, His617, Ser598, Trp620, Gly622, Asn624, Cys623, Thr591, Gly600, Tyr614, Phe370, His372, Ser588) | 4 (Ala613, Phe602, Pro133, Met135, Leu592) |
| CPT2–metformin | 12 | 3 (Phe131, Ala603, Leu129) | 8 (Asn132, Pro133, Asn130, His372, Leu343, Pro604, Val605, Ser588) | 1 (Phe602) |

Analysis of the plots revealed that against PPARA, the complex formed with phylloquinone, showed 25 interactions comprising 16 van der Waal forces (Gln277, Met330, Met320, Thr279, Gly335, Val332, Tyr334, Leu460, Tyr464, His440, Tyr314, Ser280, Thr283, Asn219, Phe318, Lys358), 1 pi–pi-T-shaped (Phe273), 6 alkyl (Met355, Ile317, Leu321, Val324, Met220, Leu331) and 2 pi–alkyl (Cys276 and Ile354) groups (Figure 6A) compared with rosiglitazone with 20 interactions (1 conventional H bond (Ser280), 14 van der Waal forces (Asn219, Phe218, Met220, Ser323, Val324, Asn221, Met320, Thr279, Phe318, Lys358, Ile354, Gln277, Thr283, Met355), 2 pi–pi-T shaped bond (His440, Phe273) and 3 pi–alkyl (Ile317, Leu321, Cys276)) (Figure 6B). With respect to FABP4, tricosylic acid and linoleic acid had comparably higher docking scores of −6.0 and −6.3 kcal/mol, respectively, though which were lower than rosiglitazone (−8.3 kcal/mol). However, the number of interactions of these SSEO compounds (24 (comprising 2 H bonds (Ala75, Thr24), 11 van der Waal forces (Glu72, Thr60, Asp76, Arg78, Arg126, Ser53, Lys58, Ser55, Val25, Tyr19, Arg106) and 11 (Met20, Ala36, Pro38, Phe51, Ala33, The16, Tyr128, Cys117, Ile104, Met40, Val115) pi–alkyl bonds) Figure 7A), and 22 (which are 1 H bond (Arg126), 12 van der Waals (Gln95, Thr74, Arg78, Val25, Asp76, Lys58, Ser53, Ser55, Cys117, Val115, Tyr128, Arg106) and 9 alkyl/ pi–alkyl (Tyr19, Phe16, Met20, Ala75, Ala33, Ala36, Pro38, Phe57, Ile104) respectively) was higher than that of rosiglitazone (22) [2 H bonds (Arg106, Ser53)], 14 van der Waal (Val23, Arg78, Tyr19, Val115, Tyr128, Ser55, Val25, Met20, Asp76, Gln95, Ala75, Ala33, Phe57, Thr60), 1 pi–cation (Arg126), 1 pi–sulfur (Phe16) and 4 pi–alkyl groups (Cys117, Ile104, Pro38, Ala36) (Figure 7B). Tricosylic acid in complexation with PPARD had the highest docking score (−7.1 kcal/mol) when compared with the other compounds; the score (which was lesser than rosiglitazone, −8.7 kcal/mol) correlated with interaction plot results, exhibiting the highest number (25) consisting of 1 H bond (His287), 9 (Phe316, Trp228, Leu317, Thr252, Phe291, Phe246, Gln250, Leu433, Met417) van der Waals, 2 unfavorable donor–donor bonds (Tyr437, Thr253), and 12 alkyl forces (Leu219, Ile213, Val312, Arg218, Val245, Leu323, Val305, Leu294, Cys249, Ile322, Ile328, Lys331, and 1 pi–alkyl force (His413) (Figure 8A). However, the highest docking value of rosiglitazone presented a lesser number of interactions (3 H bonds (Val245, His244, Ile327), 11 van der Waal forces (Leu219, Arg248, Ile328, Lys331, Met417, Phe246, Thr253, Phe291, Val305, Ile213, Trp228), 1 pi–sulfur (Cys249), 1 pi–pi-T (His413) and 3 pi–alkyl bonds (Val312, Leu317, Leu303)) (Figure 8B) compared to tricosylic acid. The PPARG–lignoceric acid complex (−6.1 kcal/mol) was best suited arising from the coming together of PPARG and lignoceric acid. The affinity of the lignoceric acid for the PPARG active site was further established by its highest number of interactions consisting of 1 H bond (Gln286), 15 van der Waals (15 (Ser289, Val339, Leu340, Ile341, Glu295, Phe226, Pro227, Phe287, Phe363, His449, Tyr327, Met364, Ser342, Lys367, Leu453) and 8 alkyl bonds (Cys285, Arg288, Leu333, Ala292, Met329, Ile326, Leu228, Leu330) as

shown in Figure 9A. While the complexes of PPARG–palmitic acid, PPARG–stearic acid, and PPARG–tricosylic acid similarly presented a higher number of interactions (20, 18, and 18, respectively), rosiglitazone had 17 interactions characterized by 6 H- bonds (Leu228, Arg288, Pro227, Phe226, Ser332, Cys285), 6 van der Waal bonds (Glu295, Ile341, Thr229, Met329, Leu333, Ser289, Ile341), 4 pi–alkyl (Ile326, Ala292, Leu330, Val339) and 1 pi–sulfur (Met364) (Figure 9B). The docking of rosiglitazone to the active site of CPT2 gave the highest docking score (−9.4 kcal/mol), indicating its higher affinity for the target; this was followed by CPT2–stearic acid with −6.3 kcal/mol, CPT2–arachidic acid (−6.2 kcal/mol), CPT2– behenic acid (−6.1 kcal/mol), and CPT2–lignoceric and tricosylic acids (−5.7 kcal/mol). However, a critical look at the result of the interaction plots presented an insight into dissimilarity in the trend of the docking score–interaction plot relationships (except rosiglitazone) as some compounds such as stearic (16 (3 H bonds, 7 van der Waals and 6 other interactions)) and arachidic [16 (2 H bonds, 9 van der Waals and 5 others) acids reflected a lower number of interactions. The CPT2–tricosylic acid system with the lowest docking value surprisingly revealed the highest number of interactions composed of 3 H bonds ((Ser488, Tyr486, Asp376), 9 van der Waals (Arg554, Trp116, Thr499, Met119, Gly377, Ser590, Asn585, Ser588, Asn130), 1 pi–sigma (Phe602), 4 alkyl (Val605, Met135, Pro133, Leu592) and 4 pi–alkyl bonds (Phe131, His372, Tyr120, Phe370) (Figure 10A) higher than rosiglitazone, which had 21 interactions (1 H bond (Ser590), 15 van der Waal forces (Val597, Ala615, Leu599, His617, Ser598, Trp620, Gly622, Asn624, Cys623, Thr591, Gly600, Tyr614, Phe370, His372, Ser588), 1 pi–pi stacked (Phe602), 1 pi–sulfur (Met135) and 3 alkyl bonds (Ala613, Pro133, Leu592)) (Figure 10B) with the highest docking score.

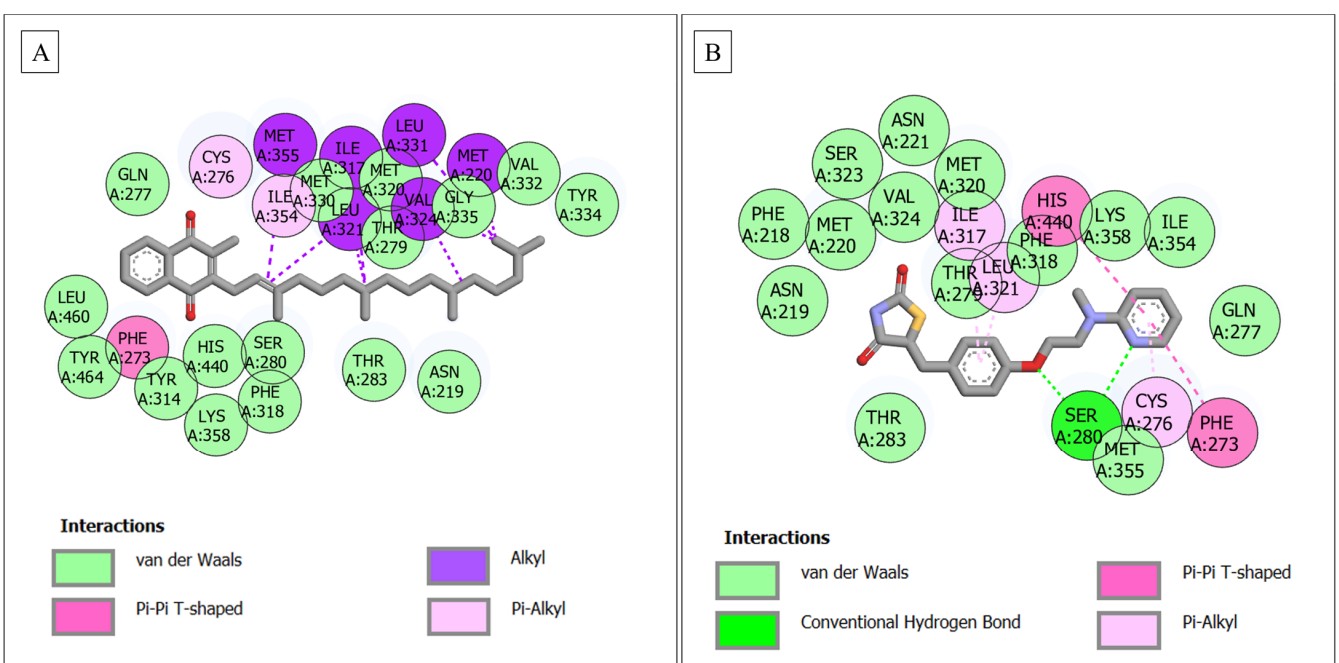

**Figure 6.** Plot of interactions: (**A**) PPARA–phylloquinone; and (**B**) PPARA–rosiglitazone over 100 ns.

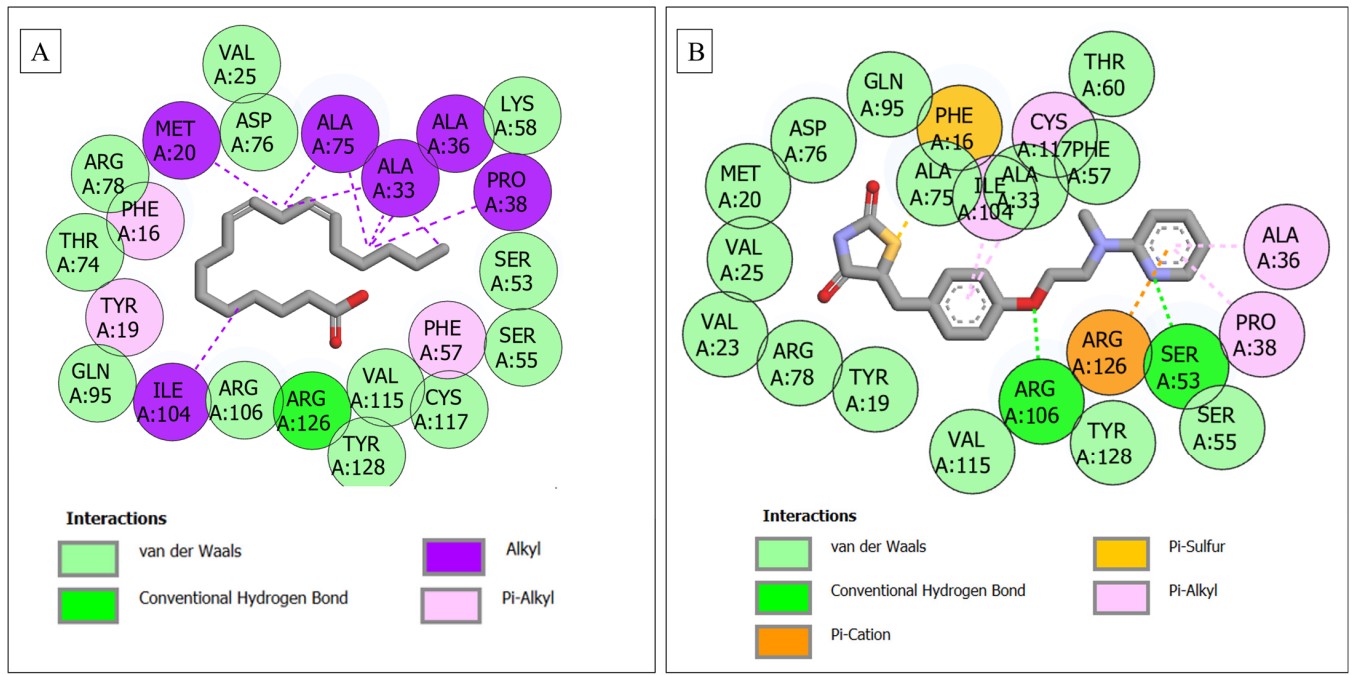

**Figure 7.** Plot of interactions: (**A**) FABP4–linoleic acid; and (**B**) FABP4–rosiglitazone.

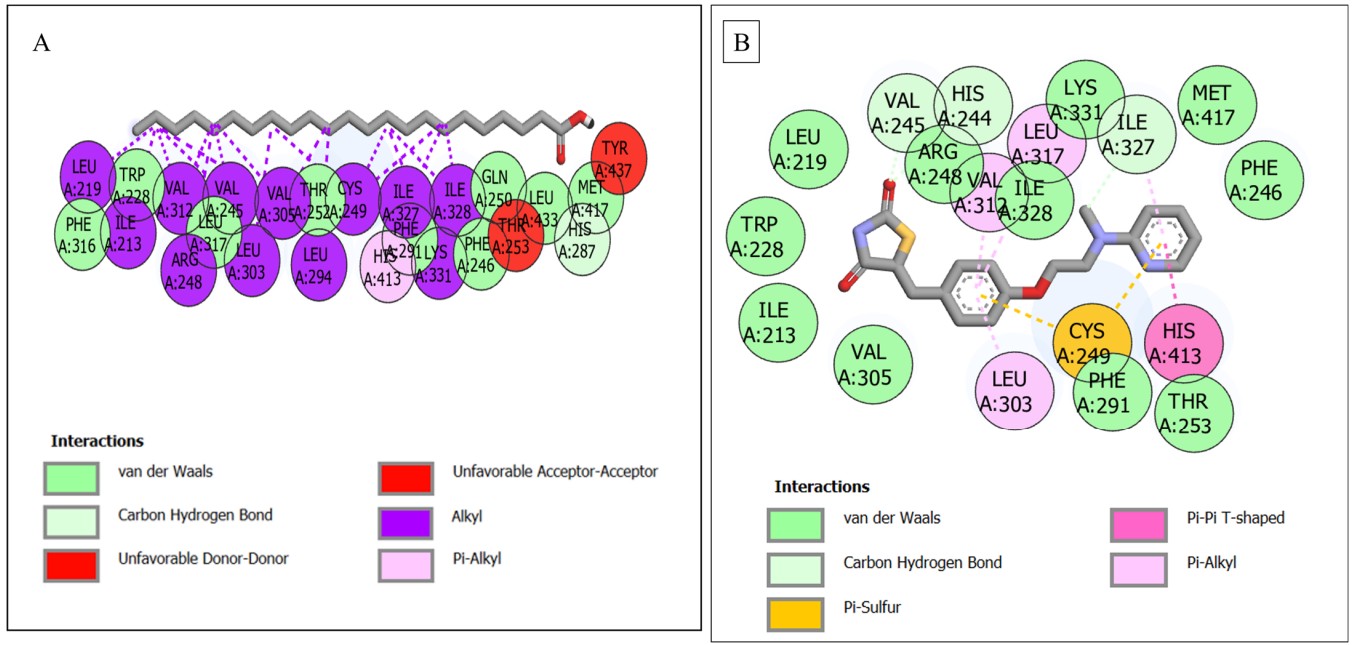

**Figure 8.** Plot of interactions: (**A**) PPARD–tricosylic acid; and (**B**) PPARD–rosiglitazone.

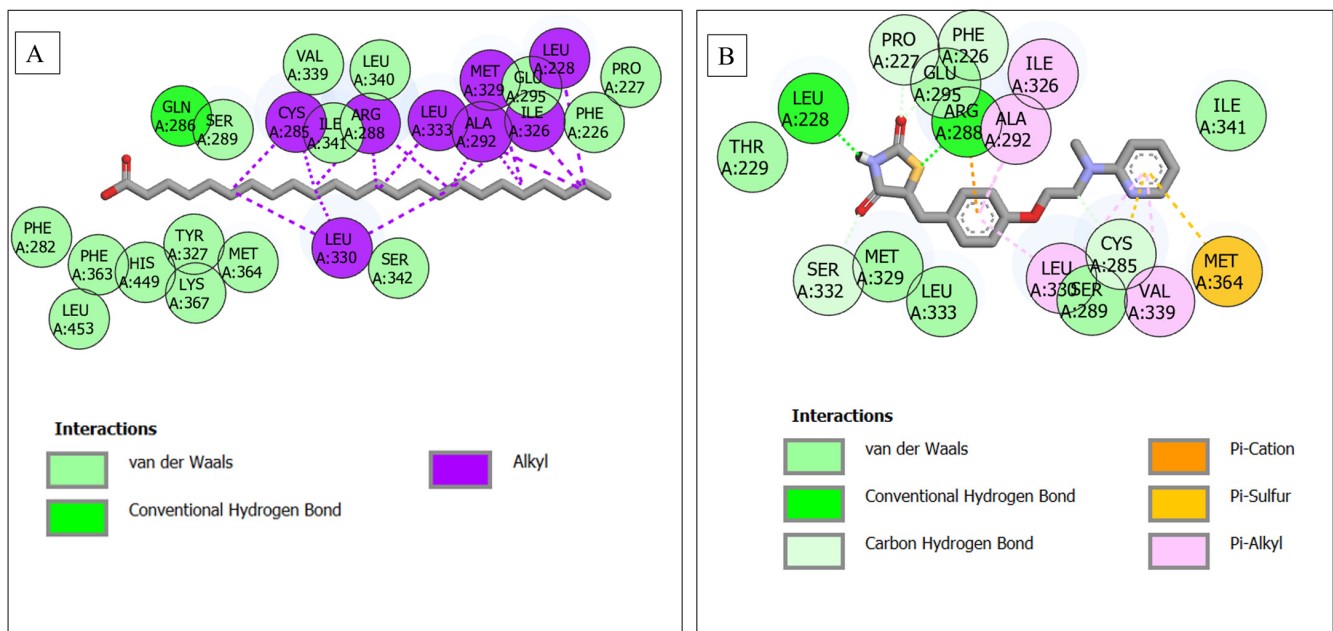

**Figure 9.** Plot of interactions: (**A**) PPARG–lignoceric acid; and (**B**) PPARG–rosiglitazone.

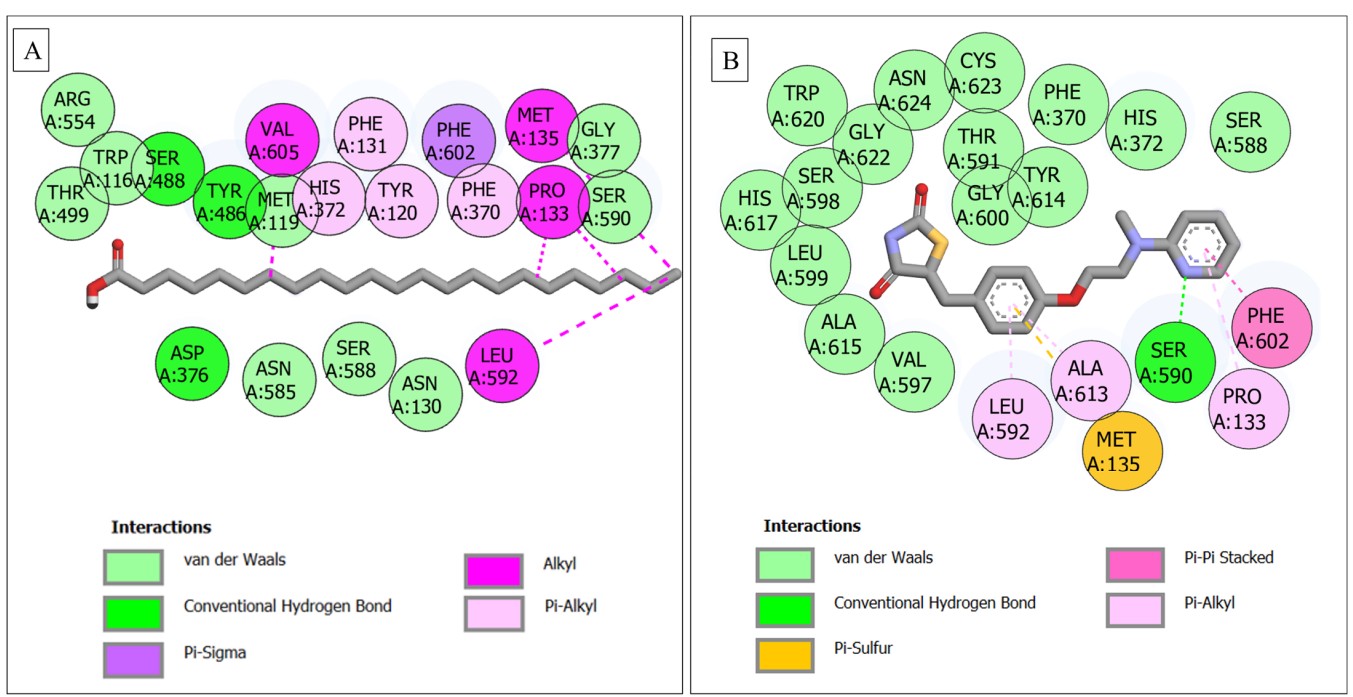

**Figure 10.** Plot of interactions: (**A**) CPT2–tricosylic acid; and (**B**) CPT2–rosiglitazone.

## 4. Discussion

The use of medicinal plants is a laudable approach and widely accepted in recent times due to its desired therapeutic effect. However, these plants containing one or more bioactive components might present a difficult task if their mechanisms of action are yet to be determined [46]. Hence, to simplify this possible complicated process, the exploration of network-pharmacology-aided molecular docking approaches may go a long way in achieving this task. The reason for this is not far-fetched, as this strategy can offer or uncover prospective drug moieties and predict gene targets and the linked signaling route of diseases, albeit infectious or non-infectious, communicable or non-communicable. It

is believed that this approach should be able to provide increased results as far as the pharmacological determination of potential plants is concerned [47].

Literature information has continued to reveal that the abundant pharmacological potential elicited by medicinal plants is partly or most times a result of the endowed phytoconstituents [48,49]. The 15 identified SSEO phytocompounds with the subsequent passing of Lipinski's rule of five (drug likeliness) with no violation suggest them as important bioactive compounds of the plant's seeds by oral route administration. The chemical profiling of the oilseed revealed unsaturated fatty acid components as predominant contributing to the quality of the oil [50], alleviating body cholesterol while offering the potential to lower the possible risk of cardiovascular diseases [51]. Besides, T2D is known to be one of the key risk factors for cardiovascular diseases [52]. Notwithstanding the aforementioned, the mechanism of antidiabetic potential of the plant or its essential oil constituents has not been elucidated to the best of our knowledge despite the established antidiabetic effect of the seeds.

The identification of 17 genes from the compound–target pathway arising from SSEO compounds and T2D targets is indicative of the involvement of many genes in the signaling pathway of T2D as expressed from the KEGG enrichment analysis establishing three metabolic pathways with only PPAR having five involvement genes specifically related to diabetes mellitus; the implication of this could be that PPAR is a germane pathway essential in the control of T2D by SSEO. It should be noted that determining the most implicated pathway is attributed to the false discovery rate (FDR), indicating that the signaling route with the lowest FDR value is the most germane route to be studied to aid in providing insights into the mechanism of action. Interestingly and notwithstanding the abovementioned, similar studies have not only identified this route as important in the downregulation of genes involved in diabetes and obesity emergence [53,54], but have also found it as key in the mechanism of the antidiabetic action of phytocompounds from berberine as well [55]. Other involved pathways are neuroactive ligand-receptor pathways and pathways in the pathogenesis of cancer; they both have been identified as important in the molecular mechanism of phytocompounds against diabetes mellitus and Alzheimer's. The former was in previous studies by Noor et al. [56,57] for *Abrus precatorius* against diabetes while the latter was studied and identified for metallothionein-III against Alzheimer's disease in a study by Roy et al. [58].

Peroxisome proliferator-activated receptors (PPARs) are transcription factor family receptors concerned with carbohydrate (glucose) metabolism. They specifically function at the DNA level to cause gene expression [59]. In fact, a report of the improved expression level of PPAR-$\gamma$-activated receptor in addition to other genes such as lipoprotein a (LPa), interleukin-1 (IL-1), and tumor necrosis factor-alpha (TNF-$\alpha$) by a related seed oil (flaxseed) have been reported [60]. Additionally, increased expression of IL-6 and TNF-$\alpha$ levels by sunflower oils has been reported in white adipose tissue and insulin-sensitive tissues [61,62]. The PPAR signaling pathway in this study expressed five relevant genes, PPARA, FABP4, PPARD, PPARG, and CPT2 which interacted well with the SSEO components (though in varying degrees). Interestingly, this route was similarly reported [54] where the molecular mechanism of action of Sorghum bicolor on T2D was studied expressing six genes including PPARA, FABP4, PPARD, and PPARG (identified in the present study) and FABP3 as well as NR1H3. Moreover, the interaction of the PPARA gene with all the 15 compounds from SSEO suggests its superiority or importance (in T2D management) among other genes and, interestingly, upregulating the expression of this gene has been reported to help in the control of the elevated level of glucose and insulin-mediated elongation of the heart cell size (cardiomyocyte hypertrophy) as well as diabetes retinopathy by related phytocompounds (berberine) [63,64]. Furthermore, PPARG and PPARD in addition to PPARA when overly expressed in adipose tissue by berberine are established to cause a reduction in glucose and lipid levels [65]. Hence, based on these submissions, it could be suggested that the mechanism of antidiabetic action of SSEO compounds could be via a modulatory role on glucose metabolism in vital body parts such as the heart and the eyes.

Molecular docking is a measure of the binding mode of a ligand to a protein [66]. Molecular docking makes use of scoring assessment as an expression of the binding mode of a ligand or inhibitor at the active site of the protein. This means that the most negative docking value will present the best orientation and inhibition of the enzyme [67]. Hence, the highest docking scores of phylloquinone (against PPARA), linoleic acid (FABP4), tricosylic acid (PPARD), lignoceric acid (PPARG), and stearic acid (CPT2) are indications of significant binding affinity towards the respective targets [68], thus promoting better complex stabilities. Consistent with observation on the binding affinity, the understanding of the kind of interaction existing between the five targets and respective phytoconstituents is key to providing information on the mechanism of action of the latter against T2D [57].

The type and number of interactions resulting in complex formation from the binding of ligand and the protein's amino acid residues is not only a consequence of eventual affinity [69], but essential in the development of a probable drug candidate [70]. A complex with a higher number of interactions often presents profound complex stability; the presence of important bonds such as H bonds and van der Waals bonds [46,71] as well as the shorter bond length existing between amino acid residues [72] are also important elements that contribute to a stronger affinity (between ligands and proteins). Intriguingly, the presence of interactions such as van der Waal forces, pi–pi stacked, and H bonds predominant between the top five active compounds and respective receptor target proteins as found in this study could be said to be indicative of their importance and possibly the well-established complex stability (of some of them) compared with to the used references. Typically, with PPARA, the higher number of interactions from the compounds as compared with the standard is an indication of their better binding affinities and correlated stabilities above the reference drug moieties. Additionally, the binding affinities of the compounds and standards depicted by the respective docking scores correlated with the established numbers of interactions.

The presence (and number) of H bonds is one of the key factors in conferring complex stability, as there are reports that H bonds contribute largely to the stability of complexes [46,67]. The lack of higher numbers of interactions of linoleic acid–FABP4 complexes, despite their high docking scores relative to tricosylic acid–FABP4, may be a result of their lower number of H bonds and other interactions (alkyl and/or pi–alkyl groups) since a higher number of H bonds contributes to the affinity and overall stability of the complex [71]. Additionally, the highest docking score of the FABP4–rosiglitazone complex presented the lowest interaction numbers compared with other SSEO compounds, possibly suggesting that the synthetic drug does not interact well enough with the amino acid residues (at the active site), particularly Arg106 and Ser53, which form two H bond interactions.

The negative docking score of the tricosylic acid–PPARD complex correlated by its highest number of interactions compared with rosiglitazone–PPARD (with lower interaction numbers) is commendable. However, the highest docking value of the latter may be said to have been contributed by the increased number of H bonds and van der Waal forces. Interestingly though, some of the other SSEO compounds have equal (pentadecylic acid (19), oleic acid (19)) or greater (stearic acid (20), arachidic acid (23)) numbers of interactions compared with rosiglitazone, which has the highest docking value. Furthermore, the high number of interactions among all the phytocompounds' complexations with PPARG compared with the antagonist and the reference standard suggested the good stabilities of these complexes. However, the docking score of rosiglitazone compared with those of the SSEO might be contributed to by the (higher number of) H bonds.

Summarily, the highest docking scores of identified compounds (phylloquinone, linoleic acid, tricosylic acid, lignoceric acid, and stearic acid) revealed the best affinities against respective targets (PPARA, FABP4, PPARD, and PPARG CPT2), as also corroborated by their high number of interactions (except stearic acid, as replaced by tricosylic acid against CPT2) in comparison with other compounds and standards, indicating their superiority. However, since the PPAR signaling pathway is concerned with diabetes and obesity emergence via the downregulation of the PPARA, FABP4, PPARD, PPARG, and CPT2

genes, and coupled with the fact that phylloquinones, linoleic acid, tricosylic acid, and lignoceric acid maintained good stabilities with these targets or genes based on molecular docking evaluation, that these four compounds could serve as probable PPAR ligands and as potential therapeutic choices against T2D, obesity, and insulin resistance [50,73] brought about by the impairment of insulin signaling [68], thereby suggesting them as probable compounds that could be further developed into drug candidates for insulin sensitization and T2D management [74]. Notwithstanding the aforementioned, the number of genes attributed to a signaling pathway is measured by its rich factor or strength [75]; thus, the higher the rich factor, the greater the degree of enrichment [68]. Intriguingly, the PPAR pathway established the highest degree of enrichment compared with other signaling routes; therefore, the present work buttresses the mechanism of action associated with the PPAR route and PPARA gene. However, this may not rule out further exploration of other identified genes. Above all, the main mechanism of the antidiabetic action of SSEO and/or its compounds may be suggested to be linked to the PPAR pathway for the regulation of glucose, as proposed in Figure 11.

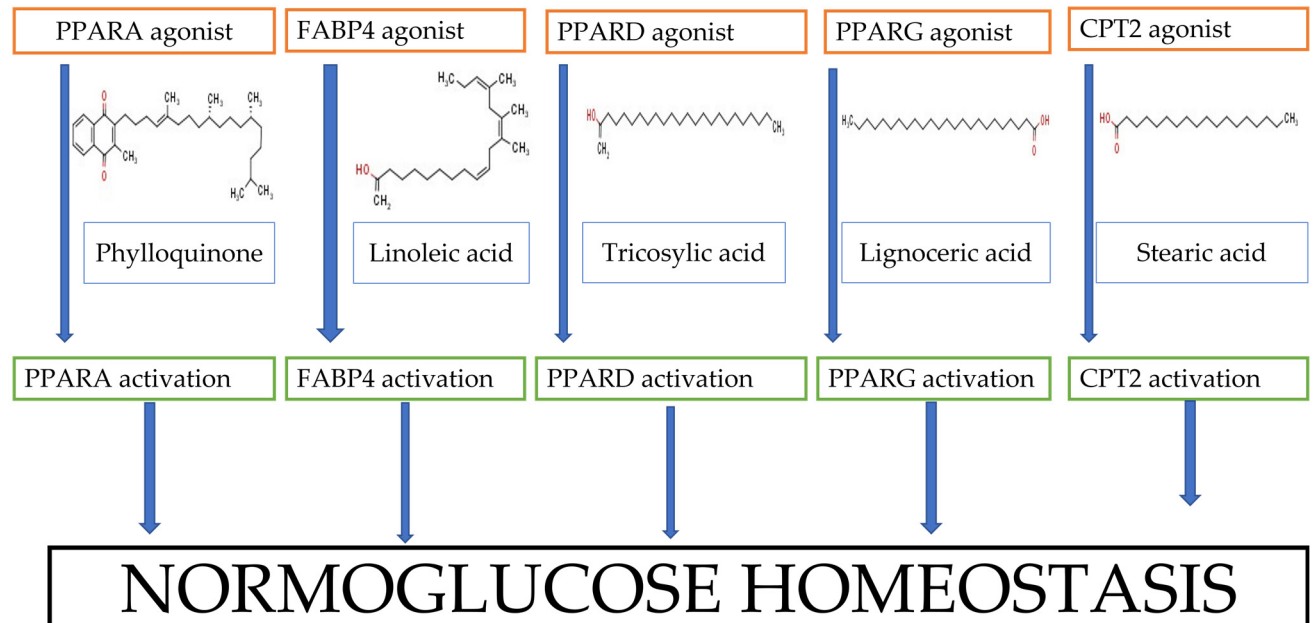

**Figure 11.** Proposed mechanism of glucose control by sunflower seed essential oil compounds.

## 5. Conclusions

The study was able to explore network-pharmacology-supported molecular docking to decipher the mechanism of the antidiabetic action of SSEO. Through NP, the study identified PPAR as the best diabetic route to underpin the intended aim and thus proposes PPARA, FABP4, PPARD, PPARG, and CPT2 as probable therapeutic targets to curb the influence or prevalence of T2D. The study concludes that while the reduction of glucose in the diabetic state may be suggested as the mechanism of antidiabetic action of sunflowers, compounds such as phylloquinone, linoleic acid, tricosylic acid, and lignoceric acid are probable drug candidates that may be further developed as effective therapeutic moieties against T2D. Further studies are encouraged to evaluate the in vitro and in vivo antidiabetic action of these phytoconstituents toward drug development.

**Supplementary Materials:** The following supporting information can be downloaded at: https: //www.mdpi.com/article/10.3390/endocrines4020026/s1. Figure S1 is the GCMS–FAMEs chromatogram identifying the sunflower seed essential oil components. Figure S2 is the chromatogram showing the retention times and $m/z$ of the compounds. Figure S3 contained the interaction plots between SSEO compounds (not presented in the text), metformin and respective genes over 100 nanosec-

onds. Supplementary Table S1 is the docking scores result of all the SSEO compounds and standards complexed with respective targets or genes (PPARA, FABP4, PPARD, PPARG and CPT2).

**Author Contributions:** Conceptualization, S.S.; methodology, A.R.; software, A.R.; validation, S.S.; writing—original draft preparation, F.O.B.; funding acquisition, S.S. All authors have read and agreed to the published version of the manuscript.

**Funding:** The financial assistance of the Directorate of Research and Postgraduate Support, Durban University of Technology, South African Medical Research Council (SA MRC) under a Self-Initiated Research Grant and NRF Research Development Grant for rated researchers (Grant number 120433) awarded to S Sabiu, are duly acknowledged.

**Institutional Review Board Statement:** Not applicable.

**Informed Consent Statement:** Not applicable.

**Data Availability Statement:** Data relating to the study are provided as supplementary files.

**Acknowledgments:** The authors appreciate the Agricultural Research Council (ARC) Grain Crop Institute, Potchefstroom, South Africa for providing the seeds used in the study and the National Research Foundation (NRF) for funding of master's program of A Rampadarath. The authors similarly acknowledge the support from NRF for the postdoctoral fellowship awarded to FO Balogun (UID: 129494) tenable at the Department of Biotechnology and Food Science, Durban University of Technology, Durban, KwaZulu-Natal, South Africa.

**Conflicts of Interest:** The authors declare no conflict of interest.

## Appendix A

1A–E. Superimposition on co-crystalized structures of [A] PPARA with ($\alpha$) the top five compounds: linoleic acid (green), lignoceric acid (yellow), behenic acid (brown), phylloquinone (cyan), and tricosylic acid (pink) with antidiabetic standard "metformin: (orange), native ligand (blue), and standard "rosiglitazone" (purple) have attained the same binding orientation as the native ligand at the active site. Root mean square deviation (RMSD) value of 0.5 Å. Grid box co-ordinates center (x = 8.74703; y = 24.7807; z = 28.7652) and size (x = 12.9933; y = 16.1297; z = 33.5524); ($\beta$) native ligand "3-{5-methoxy-1-[(4-methoxyphenyl) sulfonyl]-1H-indol-3-yl} propanoic acid" (Blue) and standard 'rosiglitazone" (purple); ($\gamma$) Best compound, phylloquinone (cyan) with native ligand (blue) at the active. [B] PPARG with ($\alpha$) the top five compounds: arachidic acid (cyan), palmitic acid (yellow), stearic acid (brown), tricosylic acid (pink), and lignoceric acid (green) with antidiabetic standard "metformin: (orange), native ligand (blue), and standard "rosiglitazone" (purple) have attained the same binding orientation as the native ligand at the active site. RMSD value of 0.5 Å. Grid box co-ordinates center (x = −5.01022; y = −2.97455; z = −25.5835) and size (x = 16.1272; y = 10.7321; z = 45.789); ($\beta$) native ligand "Tetrac" (Blue) and standard 'rosiglitazone" (purple); ($\gamma$) best compound, lignoceric acid (green) with native ligand (blue) at the active site. [C] PPARD with ($\alpha$) top five compounds: stearic acid (cyan), pentadecyclic acid (yellow), oleic acid (brown), arachidic acid (pink), and tricosylic acid (green) with antidiabetic standard "metformin: (orange), native ligand (blue) and standard "rosiglitazone" (purple) have attained the same binding orientation as the native ligand at the active site. RMSD value of 0.5 Å. Grid box co-ordinates: center (x = −6.47302; y = −18.503; z = 31.4425) and size (x = 13.7245; y = 10.304; z = 49.627); ($\beta$) native ligand "2-[2-methyl-4-[[4-methyl-2-[4-(trifluoromethyl) phenyl]-1,3-selenazol-5-yl]methylsulfanyl]phenoxy]ethanoic acid" (Blue) and standard 'rosiglitazone" (purple) at active site; ($\gamma$) best compound, tricosylic acid (green) with native ligand (blue) at the active site. [D] CPT2 with ($\alpha$) top five compounds: arachidic acid (yellow), stearic acid (pink), tricosylic acid (cyan), lignoceric acid (brown) and behenic acid (green) with antidiabetic standard "metformin: (orange), native ligand (blue) and standard "rosiglitazone" (purple) have attained the same binding orientation as the native ligand at the active site. RMSD value of 0.5 Å. Grid box co-ordinates: center (x = −14.3244; y = 8.44729; z = 38.4745) and size (x = 30.8303; y = 15.3058; z = 64.971); ($\beta$) native ligand "(3r)-3-{[(tetradecylamino)carbonyl]amino}-4-(trimethylammonio)butanoate"

(Blue) and standard 'rosiglitazone" (purple); (γ) best compound, stearic acid (pink) with native ligand (blue) at the active site. [E] FABP4 with (A) top five compounds: linoleic acid (cyan), lignoceric acid (yellow), behenic acid (green), arachidic acid (pink), and tricosylic acid (brown) with antidiabetic standard "metformin: (orange), native ligand (blue) and standard "rosiglitazone" (purple) have attained the same binding orientation as the native ligand at the active site. RMSD value of 0.5 Å. Grid box co-ordinates center (x = 4.33248; y = 7.2763; z = 12.244) and size (x = 9.84105; y = 4.26808; z = 40.406); (β) with native ligand "3-(2-phenyl-1h-indol-1-yl) propanoic acid" (Blue) and standard 'rosiglitazone' (purple); (γ) best compound, linoleic acid (cyan) with native ligand (blue).

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
