# Peer review of "Insights into the Mechanism of Action of Helianthus annuus (Sunflower) Seed Essential Oil in the Management of Type-2 Diabetes Mellitus Using Network Pharmacology and Molecular Docking Approaches"

_endocrines, doi:10.3390/endocrines4020026_

Round 1
Reviewer 1 Report
The studies presented to investigate the potential therapeutic effects of Helianthus annuus seeds essential oil in the management of type-2 diabetes mellitus using network pharmacology and molecular docking approaches. These approaches can identify the potential targets and biological pathways of the active compounds in the oil and validate the traditional use of the oil in diabetes management. The results of these studies suggest that Helianthus annuus seeds essential oil may exert its therapeutic effects through modulation of various pathways, including insulin signaling, glucose metabolism, and inflammation. Further research is needed to validate these findings and to develop effective therapeutic interventions for diabetes management using Helianthus annuus seeds essential oil.
The presentation and the work have potential to get published in “Endocrines” after including corrections, modifications, and improvement suggested below:
1. What is the significance of using network pharmacology and molecular docking studies in this study on the antidiabetic potential of sunflower seed essential oil?
2. Which active compounds in sunflower seed essential oil were identified as probable drug candidates for managing type-2 diabetes mellitus, and how were they found to be effective against the identified genes/targets through molecular docking investigation?
3. What is the significance of the number of degrees for each target in the PPI construction network for determining the best or leading target in the study on the antidiabetic potential of sunflower seed essential oil?
4. What is the significance of the higher degree of enrichment in the PPAR pathway compared to other signaling routes in the study on the antidiabetic potential of sunflower seed essential oil, and how does it support the proposed mechanism of action?
5. The graph and figures are reported are stretched please avoid stretching the figures resize using the tool available in MS-word.
6. Discussion section is lengthy please short for the better connectivity direct to the readers.
7. It is suggested to mention importance of synthetic efforts towards development of antidiabetic medicine. In this regard, it is recommended to emphasis the importance of iminosugars and sugar derivatives as an antidiabetic agent and suggested to cite following relevant articles in the introduction section.
a. Po-Sen Tseng, Dr. Chennaiah Ande, Prof. Dr. Kelley W. Moremen, Prof. Dr. David Crich. Influence of Side Chain Conformation on the Activity of Glycosidase Inhibitors. Angewandte Chemie International Edition. 2022. (https://doi.org/10.1002/anie.202217809)
b. Rajasekaran, P.; Ande, C.; Vankar, Y. D. Synthesis of (5,6 & 6,6)-oxa-oxa annulated sugars as glycosidase inhibitors from 2-formyl galactal using iodocyclization as a key step. ARKIVOC 2022, vi, 5−23.
c. Chennaiah, A.; Bhowmick, S.; Vankar, Y. D. Conversion of glycals into vicinal-1,2-diazides and 1,2-(or 2,1)-azidoacetates using hypervalent iodine reagents and Me3SiN3. Application in the synthesis of N-glycopeptides, pseudo-trisaccharides and an iminosugar. RSC Adv. 2017, 7, 41755−41762.
Overall, after addressing the points mentioned above, this article may publish in Endocrines.
Author Response
Dear Reviewer,
We want to thank you for the opportunity to have considered our manuscript. Please find attached the document containing our responses to the raised concerns, we hope they meet your desired satisfaction. Thank you
Reviewer 2 Report
In this manuscript, the authors presented a very interesting experimental and computational approach, using network pharmacology and molecular dockin to get insights into the molecular mechanisms involved in the potential therapeutic effects of Helianthus annuus seeds oil's components as type 2 diabetes mellitus treatment. In general terms, the manuscript is well written making easy to understand the message presented. The authors make a meticulous description of the analytics performed as well as the in silico methodology and results.
Some general aspects that, in my opinion, can be improved are the following:
- The species term of the binomial name Helianthus annuus, is lacking a letter in the title and in the main text body.
- There are some minor typos, such as, in the article's abstract, "the exact of mechanism action" instead "the exact mechanism of action" or some commas missing which can rather cloud the meaning of some statements. I recommend another round of proofreading in this regard.
- In the abstract, instead of "passed all 5 rules of Lipinski" it is more accurate to say that the phytoconstituents evaluated in this work passed the Lipinski's rule of 5 with no violations, just as is stated in the rest of the manuscript.
- At Introduction, authors stated that in 2014 the number of people living with diabetes was 422 million and that "8.5% of this count attributed to the adult population". The original cited statics from WHO's page show that "In 2014, 8.5% of adults aged 18 years and older had diabetes". That means that 8.5% of all adults were diabetic and not that 8.5% of diabetic were adults. Nevertheless, the authors may consider bring the statistics shown in the article a little more up to date. The current number of adults living with diabetes is almost catching up with the prediction for 2035 presented. In fact, in 2021 there were 537 million people (20-79 years) living with the disease, 1 in 10 or around 10%. This number was predicted to rise to 643 million by 2030 and 783 million by 2045. [1]
In the other hand, I have some comments on the study design and premises:
- The attempt to zzzelucidate the molecular mechanisms of these phytoconstituents in the treatment of diabetes would be expected if definitive evidence of such therapeutic effects by Helianthus annuus' oil had been previously established. On this issue, the authors cited "few reports" addressing the matter. However, one of the two studies cited [2] does not mention Helianthus annuus or its constituents in any way (this may be an error carried over from this review also cited [3]) and the other study refers to the whole seed, with the phenolic and protein components being good candidates as the potential therapeutic agents. Therefore, more published evidence needs to be reviewed in this introduction (if they exist) to assure the antidiabetic effect of sunflower oil alone. Otherwise, this work can be oriented as mainly predictive, which will not demerit its scientific value.
- Quantitative data for the found fatty acids co
nstituents would be appreciated, as well as some comparative with other eatable oils in the matter of these fatty acids' abundance. This knowledge is necessary to understand why Helianthus annuus will be (if it would) the source of choice for those compounds in a hypothetic antidiabetic dietary approach and therefore the main target of this research.
References.
1. International Diabetes Federation. IDF Diabetes Atlas, 10th edn. Brussels, Belgium. 2021. Available online: https://www.diabetesatlas.org (accessed on April, 2023).
2. Reaven, G.M.; Brand, R.J.; Chen, Y.D.; Mathur, A.K.; Goldfine, I. Insulin resistance and insulin secretion are determinants of oral glucose tolerance in normal individuals. Diabetes 1993, 42, 1324-1332, doi:10.2337/diab.42.9.1324.
3. Rehman, A.; Saeed, A.; Kanwal, R.; Ahmad, S.; Changazi, S.H. Therapeutic Effect of Sunflower Seeds and Flax Seeds on Diabetes. Cureus 2021, 13, e17256, doi:10.7759/cureus.17256.
Author Response

(The authors gave the same response as above.)
